# Induction of Robust and Specific Humoral and Cellular Immune Responses by Bovine Viral Diarrhea Virus Virus-Like Particles (BVDV-VLPs) Engineered with Baculovirus Expression Vector System

**DOI:** 10.3390/vaccines9040350

**Published:** 2021-04-06

**Authors:** Zhanhui Wang, Mengyao Liu, Haoran Zhao, Pengpeng Wang, Wenge Ma, Yunke Zhang, Wenxue Wu, Chen Peng

**Affiliations:** Key Laboratory of Animal Epidemiology and Zoonosis, College of Veterinary Medicine, China Agricultural University, Beijing 100193, China; wangzhanhui2016@163.com (Z.W.); mengyaoliu@cau.edu.cn (M.L.); zhaohrr@yeah.net (H.Z.); 18754881870@163.com (P.W.); Wenge_Ma@163.com (W.M.); zhangyk@cau.edu.cn (Y.Z.)

**Keywords:** bovine viral diarrhea virus, virus-like particles, baculovirus expression vector system, E^rns^, E2, vaccine

## Abstract

Bovine viral diarrhea virus (BVDV) is an important animal pathogen that affects cattle. Infections caused by the virus have resulted in substantial economic losses and outbreaks of BVDV are reported globally. Virus-like particles (VLPs) are promising vaccine technology largely due to their safety and strong ability to elicit robust immune responses. In this study, we developed a strategy to generate BVDV-VLPs using a baculovirus expression vector system (BEVS). We were able to assemble BVDV-VLPs composed of dimerized viral proteins E2 and E^rns^, and the VLPs were spherical particles with the diameters of about 50 nm. Mice immunized with 15 μg of VLPs adjuvanted with ISA201 elicited higher levels of E2-specific IgG, IgG1, and IgG2a antibodies as well as higher BVDV-neutralizing activity in comparison with controls. Re-stimulation of the splenocytes collected from mice immunized with VLPs led to significantly increased levels of CD3^+^CD4^+^T cells and CD3^+^CD8^+^T cells. In addition, the splenocytes showed dramatically enhanced proliferation and the secretion of Th1-associated IFN-γ and Th2-associated IL-4 compared to that of the unstimulated control group. Taken together, our data indicate that BVDV-VLPs efficiently induced BVDV-specific humoral and cellular immune responses in mice, showing a promising potential of developing BVDV-VLP-based vaccines for the prevention of BVDV infections.

## 1. Introduction

Bovine viral diarrhea virus (BVDV) is an important pathogen of cattle found in many parts of the world, which poses a great threat to agriculture globally. BVDV is capable of infecting a diverse range of animals, including pigs, sheep, goats, deer, and camelids [1]. Upon infection, BVDV often induces both acute infections (AI) and persistent infections (PI) in cattle. Symptoms associated with AI include diarrhea, fever, leukopenia, coughing, and increased nasal discharge. PI is established when a non-cytopathic (NCP) BVDV crosses the barrier of placenta and infects an immunoincompetent fetus. Persistently infected calves are the major source to spread BVDV as they can shed virus throughout their lives, and viruses can be detected in almost all organs [2,3] from these calves.

BVDV is an enveloped, positively-stranded RNA virus that belongs to the genus *Pestivirus* in the *Flaviviridae* family [4]. BVDV is classified into three genotypes: BVDV-1(BVDV-1a~BVDV-1u), BVDV-2(BVDV-2a~BVDV-2d), and BVDV-3 (Hobi-like, atypical pestivirus) [5,6]. On the basis of the cytotoxicity in cell culture, each BVDV strain has two biotypes: NCP and cytopathic (CP) [7]. The genome of BVDV is about 12.3 kb, which is composed of a 5’ untranslated region (UTR) containing a highly conserved internal ribosome entry site (IRES), a 3’ non-coding region (NCR), and an open reading frame (ORF) encoding a 3988-amino acid polyprotein [8]. The fully translated polyprotein is processed by cellular and viral proteases to generate 11 functional proteins, namely, NH2-Npro (N-terminal autoprotease), C (capsid protein, core), E^rns^ (envelope protein RNase secreted), E1, E2, p7, NS2-3 (NS2 and NS3), NS4A, NS4B, NS5A, and NS5B [9,10].

The two surface proteins E^rns^ and E2 are highly glycosylated and often exist as homodimers mediated by disulfide bonds. Specifically, E^rns^ contains 8–9 conserved cysteine residues that form intra- and inter-molecular disulfide bonds, more than 50% carbohydrates in the mature form of E^rns^ [11,12,13,14]. E2 has 3–6 N-linked glycosylation sites and 15–17 cysteine residues that are conserved across all genotypes [15,16]. Moreover, E2 is a membrane-anchored type I transmembrane protein with an N-terminal ectodomain and a C-terminal hydrophobic anchor [17]. BVDV’s entry into host cells is mediated by E2, which binds the cell-surface receptor CD46 [18]. E2 and E^rns^ are the main targets for neutralizing antibodies induced by BVDV infection, making them important subunit antigen candidates for vaccine development. 

Virus-like particles (VLPs) are non-infectious and genome-free virus particles constructed by one or multiple viral proteins. VLPs are similar to infectious virions in structure and conformation, but are non-infectious due to the lack of genetic material. Compared to individual proteins or peptides, VLPs display more repetitive epitopes on the surface, which may trigger stronger B cell and T cell-mediated immune responses [19,20].

Baculovirus expression vector system (BEVS) has been widely used in the production of VLPs and has been used for the development of several licensed vaccines, such as HPV16/18 vaccine (CERVARIX, GSK, Brentford, UK) and influenza virus vaccine (NanoFlu, Novavax, Gaithersburg, MD, USA) [21,22,23]. Baculovirus has a large capacity for the incorporation of foreign genes, infects only arthropods, and is essentially nonpathogenic to mammals. Moreover, baculovirus shows a strong adjuvant activity [24,25,26,27], which may help improve the immunogenicity of VLP-based vaccines. The availability of cell lines suitable for suspension cultures in serum-free conditions allows for the large-scale production of recombinant proteins. Importantly, most of the proteins expressed in BEVS undergo post-translational modifications, such as N-glycosylation, O-glycosylation, or phosphorylation, which are important for maintaining the immunogenicity of recombinant antigens [28].

BEVS-based VLPs have been successfully generated for many members of the *Flaviviridae* family, including dengue fever virus (DENV) [29], Japanese encephalitis virus (JEV) [30], Zika virus (ZV) [31], West Nile virus (WNV) [32], and hepatitis C virus (HCV) [33]. However, there has been no reports of BEVS-based VLPs for BVDV, another member of the *Flaviviridae* family. In this study, we generated BVDV-VLPs containing E^rns^ and E2 with BEVS method. We observed the assembly and structure of the VLPs with transmission electron microscopy (TEM) and immune-electron microscopy (IEM). In addition, we immunized BALB/c mice with the BVDV-VLPs we manufactured and evaluated and compared the humoral and cellular immune responses in the animals to that of a commercial BVDV vaccine. Our results suggest BVDV-VLPs generated by BEVS are a promising candidate vaccine for the prevention of BVDV infections.

## 2. Materials and Methods

### 2.1. Cells and Virus

MDBK cells, BVDV 1a strain NADL, were obtained from the China Veterinary Culture Collection Center (CVCC, Beijing, China). Sf9 (Invitrogen, Carlsbad, CA, USA) cells were cultured in suspension in the serum-free Grace medium (Invitrogen, Carlsbad, CA, USA) supplemented with 1% penicillin–streptomycin (Solarbio, Beijing, China) at 27 °C, 100 rpm. MDBK cells were maintained in Dulbecco’s Modified Eagle Medium (DMEM) (Invitrogen, Carlsbad, CA, USA) supplemented with 10% fetal bovine serum (Biological industries, Beit-Haemek, Israel) and 1% penicillin–streptomycin (Sigma-Aldrich, St. Louis, MO, USA) at 37 °C and 5% CO_2_.

### 2.2. Construction of Recombinant Baculovirus

The coding sequences of the E^rns^ proteins (residues 271-497) and E2 proteins (residues 693-1036) were cloned from BVDV 1a NADL (GenBank accession no. AJ133738.1), a gp64 signal peptide sequence (gtaagcgctattgttttatatgtgcttttggcggcggcggcgcattctgcctttgcg) was fused to the N-terminal of E^rns^ or E2 with a Gly-Gly-Gly-Gly-Ser linker, and gp64-E^rns^ and gp64-E2 were cloned into the XhoI-KpnI and BamHI-PstI sites of pFastBac Dual vector (Invitrogen, Carlsbad, CA, USA). The correct clones were verified via Sanger sequencing and were named pFastBac-E^rns^+E2, which was transformed into DH10Bac *Escherichia coli* (*E. coli*) cells to generate recombinant bacmids rBacmids-E^rns^+E2. Of the Sf9 cells, 10^6^ were transfected with 1μg rBacmids-E^rns^+E2 using Cellfectin II (Invitrogen, Carlsbad, CA, USA) according to the manufacturer’s guidelines. After 5 days, medium from cell culture was collected, centrifuged at 1000× *g* for 5 min to remove cellular debris, and stored at −80 °C.

### 2.3. Production and Purification of VLPs

Sf9 cells at a density of 3 × 10^6^ cells/mL were infected with P4 (passage #4) rBac-E^rns^+E2 with the multiplicity of infection (MOI) of 0.1 for 5 days at 27 °C, and rotated at 100 rpm in a 125 mL culture bottle. Cell suspensions were centrifuged at 1000× *g* for 5 min to remove supernatant, and the cell pellets were resuspended with 5× volumes of TNE buffer (20 mM Tris-HCl (pH 8.0), 150 mM NaCl, 2 mM EDTA) and sonicated 5 times for 10 s on ice with a sonicator (Scientz, Ningbo, China) at 30% amplification. Cell lysates were collected after centrifugation at 12,000 rpm for 20 min and used in the subsequent experiments.

For VLP purification, the cell lysates were loaded onto gradient sucrose solution (50%, 40%, 30%, 20%, and 10%, from densest to lightest using 2 mL per layer) in TNE buffer in a 13.3 mL ultraclear centrifuge tube (Beckman Coulter, Brea, CA, USA) and centrifuged at 35,000 rpm, 4 °C for 4 h in an SW41 rotor (Beckman Coulter, Brea, CA, USA). Fractions of 1.5 mL were collected dropwise after puncturing the bottom of the tube. The distribution of the VLPs in the gradient was measured by SDS-PAGE (Bio-Rad, Hercules, CA, USA) with Coomassie Brilliant Blue staining and Western blot with E^rns^ polyclonal antibody (pAb) (prepared by our laboratory) and E2 monoclonal antibodies (mAb) 348 (VMRD, Pullman, WA, USA).

Several fractions of the sucrose gradient containing the VLPs were mixed with 2× volumes of TNE and centrifuged at 35,000 rpm, 4 °C, for 2 h in an SW41 rotor (Beckman Coulter, Brea, CA, USA). VLP pellets were resuspended in 500 μL of TNE and stored at −80 °C.

### 2.4. Western Blotting Analyses

Purified VLPs or cell lysates were separated by SDS-PAGE under reducing (with 4% β-mercaptoethanol) or non-reducing conditions (without β-mercaptoethanol), and proteins were transferred to PVDF membranes (Millipore, Billerica, MA, USA) and blocked for 2 h at room temperature with 5% skim milk in PBST (0.05% Tween in 0.01 M PBS). Blocked membranes were then incubated with primary antibodies E^rns^ pAb and E2 mAb 348 diluted in 5% skim milk in PBST for 1 h at 25 °C, washed 3 times with PBST, and then incubated with HRP-conjugated secondary antibody in 5% skim milk in PBST for 30 min at 25 ℃. Membranes were washed extensively, and signals were detected using the ECL reagents (Tanon, Shanghai, China).

### 2.5. TEM and IEM

Sf9 cells infected by rBac-E^rns^+E2 were spun down and resuspended in 2.5% glutaraldehyde, followed by dehydration using a standard graded series of acetone solutions including 50%, 70%, 90%, and 100% for 15 min per step. Cells were then embedded in Epon815 resin (Sigma-Aldrich, St. Louis, MO, USA) and allowed to polymerize for 24 h at 37 °C, 24 h at 45 °C, and 24 h at 60 °C. Ultrathin sections (70 nm) of these blocks were obtained with a Reichert-Jung Ultracut E ultramicrotome, and sections were deposited on carbon-coated Cu grids and negative stained with 5% uranyl acetate and 5% lead citrate. Specimens were viewed with a JEM-1200EX (JEOL, Tokyo, Japan) transmission electron microscope at 80 kV.

For visualization of the purified VLPs, we applied 20 μL of VLPs onto carbon-coated Cu grids, which were negative stained with 2% uranyl acetate and viewed with the method described above.

For IEM, 20 μL of VLPs were applied onto carbon-coated Cu grids and blocked with 1% BSA. The primary antibodies were E^rns^ pAb and E2 mAb 348. The 10 nm colloidal gold-conjugated goat anti-rabbit antibody and 5 nm colloidal gold-conjugated goat anti-mouse antibody (BB International, Cardiff, UK) were used as the secondary antibody, and subsequently the grids were negatively stained with 2% uranyl acetate and examined with a HT7800 (Hitachi, Tokyo, Japan) transmission electron microscope at 80 kV.

### 2.6. Immunization of Mice

A total of 48 6–8-week-old female BALB/c mice (Charles River Laboratories, Beijing, China) were randomly divided into 8 groups (6 per group) and immunized intramuscularly on days 0 and 21 with (I) 5 µg of VLPs, (II) 5 µg of VLPs adjuvanted with 5 µg of ISA201, (III) 10 µg of VLPs, (IV) 10 µg of VLPs adjuvanted with 10 µg of ISA201, (V) 15 µg of VLPs, (VI) 15 µg of VLPs adjuvanted with 15 µg of ISA201, (VII) commercial BVDV inactivated vaccine, and (VIII) TNE and equal volume ISA201.

Blood samples were taken through the inner canthal orbital vein for serological analysis on day 42 post-prime immunization, and spleens were harvested from groups (V), (VI), (VII), and (VIII) for flow cytometry and splenocytes proliferation test on day 28 post-prime immunization.

### 2.7. ELISA

BVDV E2-specific serum antibody titers were measured by ELISA. We coated 96-well microtiter plates with 100μL per well of 4μg/mL of purified prokaryotic expressed BVDV-E2 protein produced in *E. coli* using a pMAL-c5X-His expression vector in CBS (0.015 M Na_2_CO_3_, 0.035 M NaHCO_3_, pH 9.6) overnight at 4 °C. Next, the plates were blocked with 5% skim milk in PBST (0.05%Tween in 0.01 M PBS) for 2 h at 37 °C. A total of 100 μL of serum samples were diluted 50x times in PBST and added to each well in the plates for 1 h at 37 °C. The plates were then washed 3 times with PBST, and 100 μL of HRP-conjugated goat anti-mice IgG (Solarbio, Beijing, China), IgG1 (abclonal, Wuhan, China) or IgG2a (Sigma-Aldrich, St. Louis, MO, USA) were 2000-, 4000-, or 500-fold diluted, respectively, in PBST and added to the plates for 1 h at 37 °C. The plates were washed again and 100 μL of TMB (Solarbio, Beijing, China) was added to each well prior to adding 50 μL of stopping solution (2 M H_2_SO_4_). OD_450nm_ was determined using an ELISA plate reader (Tecan, Männedorf, Switzerland).

### 2.8. Neutralizing Antibody Assay

The serum samples were heat-inactivated at 56 °C for 30 min and serially diluted twofold in DMEM medium, and then an equal volume of DMEM-diluted BVDV virus containing 100 TCID_50_ was added and samples were incubated for 1 h at 37 °C in a final volume of 200 μL. MDBK cells were infected in triplicate with 100 μL of the neutralization mixture and incubated for 1 h at 37 °C with 5% CO_2_, the mixture was then removed and replaced with DMEM medium with 2% FBS. Cells were further incubated for 7 days at 37 °C with 5% CO_2_, and cytopathic effects (CPE) were observed. The 50% neutralization titer of serum was calculated by the Reed and Muench method.

### 2.9. Flow Cytometry

Spleens were collected from mice on day 28 post-prime immunization and were grounded in a 35 mm Petri dish containing 5 mL of mice 1× lymphocyte separation medium (DAKEWE, Shenzhen, China); suspensions were filtered using 40-μm filters into a new tube and were centrifuged at 800× *g* for 30 min. The lymphocyte layer was collected and washed with RPMI 1640 medium(Invitrogen, Carlsbad, CA, USA), and cell counts were stained with 0.4% trypan blue and counted using the Countess Automated Cell Counter system (Life Technologies, Carlsbad, CA, USA). Splenocytes were used for subsequent flow cytometry analysis, enzyme-linked immunospot assay (Elispot), and splenocyte proliferation assays. For flow cytometry, splenocytes (5 × 10^6^ cells in 6-well cell culture plates) were stimulated with 10 μg/mL VLPs or inactivated BVDV in the presence of 3 μg/mL Brefeldin A (eBioscience, San Diego, CA, USA) for 6 h at 37 °C and 5% CO_2_, and cells were washed using staining buffer (eBioscience, San Diego, CA, USA) and stained with PE/cyanine7 anti-mice CD3 (BioLegend, San Diego, CA, USA), PE anti-mice CD4 (BioLegend, San Diego, CA, USA), and PerCP/cyanine5.5 anti-mice CD8a (BioLegend, San Diego, CA, USA). Cells were analyzed using BD Fortessa (BD Biosciences, CA, USA).

### 2.10. MTT Assay

Splenocytes (10^6^ cells per well) were added to 96-well plates with 1 μg/mL of VLPs or heat-inactivated BVDV, and splenocytes cultured with RPMI 1640 medium alone or 1 μg/mL of ConA (Sigma-Aldrich, St. Louis, MO, USA) were used as negative and positive controls, respectively. The plates were incubated at 37 °C, 5% CO_2_, for 72 h. Then, 500 μg/mL of MTT (Sigma-Aldrich, St. Louis, MO, USA) solution was added to each well and the plates were incubated at 37 °C, 5% CO_2_, for 4 h. The MTT solution was removed and 150 μL of DMSO was added to dissolve the precipitate at 37 °C for 10 min; the OD_570nm_ was determined using an ELISA plate reader (Tecan, Männedorf, Switzerland). The stimulation index (SI) was calculated as follows: SI = OD_570nm_ (sample)/OD_570nm_ (blank control).

### 2.11. EliSpot

Mouse IFN-γ precoated ELISpot kit and Mouse IL-4 precoated ELISpot kit (DAKEWE, Shenzhen, China) were used to determine cytokine expression. Splenocytes (10^6^ cells per well) were added to anti-mouse IFN-γ or IL-4 monoclonal antibody-precoated ELISpot plates with 10 μg/mL VLPs or inactivated BVDV; splenocytes cultured with RPMI 1640 medium alone or 50 μg/mL PMA/ionomycin were used as negative and positive controls, respectively. The plates were incubated at 37 °C, 5% CO_2_, for 48 h. Then, the blots were immunostained according to the manufacturer’s instructions and counted using an AID EliSpot Reader (Autoimmun Diagnostika, Strassberg, Germany). Results are presented as mean number of IFN-γ- or IL-4-secreting cells per 10^6^ splenocytes.

### 2.12. Statistical Analysis

Results are expressed as mean ± standard deviation (SD). Statistical analysis was performed by one-way analysis of variance (ANOVA) with Tukey’s multiple-comparison test in GraphPad Prism.

## 3. Results

### 3.1. Expression of E^rns^ and E2 Proteins in Sf9 Cells

The baculovirus gp64 signal peptide was fused to the N-terminal ends of E^rns^ and E2; the gp64-E^rns^ and gp64-E2 were then inserted into pFastBac Dual vector to generate the recombinant plasmid pFastBac-E^rns^+E2 and the subsequent recombinant bacmid rBacmids-E^rns^+E2. The bacmid was transfected into Sf9 cells and the recombinant baculoviruses rBac-E^rns^+E2 were rescued using a previously described protocol (Figure 1A and Appendix A). Sf9 cells infected with rBac-E^rns^+E2 typically display the following characteristics: enlargement of cell and nucleus, rounding of infected cells, and cell detachment and lysis. Next, rescued viruses were passaged 3x times for amplification. Sf9 cells were then infected with the recombinant viruses and cell lysates were collected 5 days post-infection (dpi) and proteins were separated using SDS-PAGE. Expressions of E^rns^ and E2 were examined by Western blot analyses. As shown in Figure 1B, expression of both E^rns^ and E2 proteins were detected in Sf9 cells.

### 3.2. Identification of Particle Formation in Sf9 Cells

To determine whether E^rns^ and E2 formed VLPs, we examined ultrathin section of Sf9 cells infected with rBac-E^rns^+E2 by TEM, as shown in Figure 2A. Spherically shaped VLPs stacked in intracellular vesicles were observed in cells, and the majority of the particles showed diameters of about 50 nm; similar to wild-type BVDV particles, these objects were absent from uninfected Sf9 cells.

### 3.3. Production, Purification, and Characterization of the BVDV-VLPs

In order to purify VLPs, we harvested cell lysates from Sf9 cells infected with rBac-E^rns^+E2 and placed them on top of a sucrose density gradient. Samples were centrifuged at 35,000 rpm for 4 h and then nine fractions (1.5 mL each) were collected from bottom to top. The fractions were next analyzed by SDS-PAGE and total proteins were visualized by Coomassie Brilliant Blue staining. Expression of E2 and E^rns^ were examined by Western blotting analysis. As shown in Figure 3A, two protein bands migrating at the expected size of ≈48 kDa were visible in fractions 4–6 on the Coomassie-stained gel, and Western blot result confirmed that the bands were indeed E^rns^ and E2 (Figure 3B and Appendix A).

An alternative approach was taken to confirm the identity of the protein bands collected from fractions 4–6. Specifically, contents from fractions 4–6 were purified one more time with ultracentrifugation prior to analyzing with liquid chromatography tandem mass spectrometry (LC–MS/MS) (Agilent Technologies, Palo Alto, CA, USA). The results were compared against the profile of BVDV NADL (Figure 3C). Using this method, we were able to confirm that the two bands were indeed BVDV E^rns^ and E2.

### 3.4. Observation of VLPs by TEM and IEM

To confirm the quality and purity of VLPs, we analyzed the sucrose-purified fraction 4–6 containing the highest E^rns^ and E2 concentration using TEM. TEM data showed that the VLPs were spherical in morphology, with diameters of about 50 nm (Figure 4). IEM using E^rns^ pAb and E2 mAb 348 further indicated the exposure of E^rns^ and E2 on the outer surface of the BVDV-VLPs. These data demonstrate that BVDV-VLPs comprised E^rns^ and E2 proteins (Figure 4B–D).

### 3.5. Homodimerisation of E^rns^ or E2 Proteins in VLPs

As native E^rns^ and E2 often form homodimers in BVDV-infected cells, we next sought to investigate if they also form homodimers in VLPs. Cell lysates were separated with SDS-PAGE under reducing (R) or nonreducing (NR) conditions and Western blot was performed to determine the expression and multimerization of E^rns^ and E2. As shown in Figure 5 and Appendix A, for both E2 and E^rns^, only one predominant band of 48 kDa was detected under reducing condition. As a comparison, only one band migrating at the size of 96 kDa was detectable under non-reducing condition, suggesting the existence of homodimers for both proteins.

### 3.6. Immunization With BVDV-VLP-Induced E2-Specific IgG, IgG1, IgG2a, and BVDV-Neutralizing Antibodies

To determine the immunogenicity of BVDV-VLPs, we used three doses of VLPs with or without ISA201 adjuvant to immunize BALB/C mice; a commercial BVDV inactivated vaccine and ISA201 adjuvant alone were included as positive and negative controls, respectively. Serum was collected on day 42 post-prime immunization and BVDV E2-specific IgG, IgG1, and IgG2a were examined by ELISA. As shown in Figure 6A–C, inactivated BVDV vaccine triggered strong induction of IgG, IgG1, and IgG2a in mice after immunization, validating the specificity and effectiveness of this assay. The VLPs induced IgG, IgG1, and IgG2a in a dose-dependent and adjuvant-dependent manner. VLPs alone induced low levels of IgG, IgG1, and IgG2a, however, when used with adjuvant, the VLPs resulted in dramatic induction of immunoglobulins to levels comparable to that induced by the inactivated vaccine.

To determine if the antibodies elicited by BVDV-VLPs exhibited neutralizing potentials, we collected serum from the vaccinated animals on day 42 post-prime immunization, heat-inactivated the serum at 56 °C for 30 min, and determined their capacity to neutralize BVDV infection in MDBK cells. As shown in Figure 6D, while the adjuvant alone did not elicit any neutralizing titers, the commercial vaccine induced a high level of neutralizing capability. Importantly, the BVDV-VLPs at three different doses all evoked strong neutralizing activities in mice and the titers were comparable to that induced by the commercial BVDV vaccine.

### 3.7. Activation of T cells by BVDV-VLPs

To determine if VLPs could also induce T cell immune responses, we collected mice splenocytes on day 28 post-prime immunization from animals immunized and re-stimulated with 10 μg/mL of VLPs or inactivated BVDV. The abundances of CD3^+^CD4^+^T cells and CD3^+^CD8^+^T cells were analyzed by flow cytometry. As shown in Figure 7, the ratio of CD3^+^CD4^+^T cells and CD3^+^CD8^+^T cells were significantly higher in animals treated with VLPs than those treated with the adjuvant alone or the commercial vaccine, suggesting the advantage of BVDV-VLPs in the stimulation of T cell responses over the commercial vaccine.

### 3.8. The Effect of VLPs on the Proliferation of Splenocytes

Proliferation of splenocytes upon vaccination signifies a robust activation of the immune responses. To explore the effect of BVDV-VLPs on the proliferation of splenocytes, we harvested splenocytes from immunized mice and performed MTT assay to monitor cell proliferation. As shown in Figure 8, VLPs triggered significantly enhanced proliferation of splenocytes as compared to the adjuvant alone, while the commercial vaccine failed to do so. The treatment of ConA, a commonly used mitogen, was able to further enhance cell proliferation, validating the effectiveness of the assay.

### 3.9. The Effect of VLPs on Cytokine Expression

As IL-4 and IFN-γ are major cytokines regulating antibody class switching during BVDV infections, as reported previously, we decided to monitor these two cytokiens following the treatment of VLPs in mice. We chose to monitor Th2-associated IL-4 and Th1-associated IFN-γ in spleen with EliSpot assay. Splenocytes collected from the immunized animals were stimulated with the same antigen as that used for immunization, and the number of IL-4- or IFN-γ-secreting cells were measured and plotted. Remarkably, VLPs were able to trigger improved IL4 and IFN-γ production. In comparison, the commercial vaccine was only able to enhance IFN-γ production up to 50% of the capacity induced by VLPs, and was incapable of inducing any IL-4 production (Figure 9).

## 4. Discussion

BVDV is a notorious pathogen in the dairy industry that affects farm animals around the world. In China, the seroprevalence of BVDV in dairy cattle was found to be 57%, and positive rate of viral RNA was estimated at 27.1% according to a meta-analysis performed between 2003 and 2009, and between 2010 and 2018 [34]. An independent study showed the RNA-positive rate of the virus in bulk tank milk was 43.7%, and BVDV-1a, 1c, and 1m were the dominant strains identified [35]. BVDV is the most critical threat to the cattle industry, as symptoms of BVDV infection, such as abortion, diarrhea, and embryonic death, highly jeopardize animal growth and milk production [36]. Furthermore, BVDV causes tremendous damage to the host’s immune system, and BVDV-mediated immunosuppression renders the cattle more susceptible to other infectious diseases.

Currently, both inactivated and live attenuated vaccines are commercially available for BVDV. However, drawbacks are associated with both types of vaccines. The attenuated vaccine could harm pregnant cows by causing premature miscarriage, fetal damage, or PI. PI animals are the main source of BVDV transmission. On the other hand, inactivated vaccines are safer for animals but induce a much weaker cellular immune response as compared to the attenuated vaccine. In addition, the vaccinated animals cannot be serologically distinguished from those infected with the wild-type virus, posing a challenge for viral surveillance in the cattle industry. As a result, more efforts are needed to explore the options of safe and effective BVDV vaccines.

E^rns^ and E2 are the most immunogenic proteins of BVDV and induce high titers of protective neutralizing antibodies after infection; several studies based on vaccination with E^rns^ and E2 glycoprotein expressed in different systems have been reported [37,38,39,40,41], Chimeno Zoth S used BEVS to express the E2 protein of the BVDV NADL strain, proving that the recombinant E2 protein vaccine can elicit an in vitro neutralizing humoral immune response, not only against the homologous strain, but also against heterologous BVDV strains [37]. As viral envelope glycoproteins, E^rns^ and E2 need to undergo certain N-linked glycosylation to help them fold correctly [42]. BEVS has a complete post-translational modification function.

In this study, we aimed to develop a new type of vaccine by constructing VLPs with viral E^rns^ and E2 proteins using the previously established BEVS method. A comprehensive protein domain analysis by TMHMM suggests that E2 contains an extracellular domain (amino acid residues 693–1036), a transmembrane domain (residues 1037–1059), and an intracellular domain (residues 1060–1066). E^rns^, on the other hand, does not contain any transmembrane domain. Neither E^rns^ nor E2 contain any signal peptide. Therefore, a signal peptide derived from the baculovirus gp64 surface glycoprotein was fused to the N-termini of E^rns^ and E2 extracellular domains. The baculovirus gp64 signal peptide contains hydrophobic amino acids, which is recognizable by the receptor on the endoplasmic reticulum [43]. The nascent E^rns^ and E2 enter the ER, assemble into VLPs, and are released from the ER [44].

We verified the morphology of VLPs produced by TEM. As expected, a large number of VLPs were observed in the cytoplasm; most importantly, the size and morphology of VLPs are similar to that of BVDV (Figure 2). Next, sucrose density gradient centrifugation was employed to purify VLPs as previously described [45]. The purified VLPs were characterized by TEM and IEM (Figure 4), and the content of the VLPs was examined and verified by SDS-PAGE, Western blot, and LC–MS/MS. Two bands corresponding to E^rns^ and E2 were clearly visible on the SDS-PAGE gel, and the results of LC–MS/MS confirmed that two protein bands were BVDV E^rns^ and E2 (Figure 3). Importantly, we were able to validate that both E2 and E^rns^ form homodimers (Figure 5), which was reported to be the native form for both in virus particles [46,47].

E^rns^ forms homodimers through disulfide bonds formed between cysteine 171 (C171), which is conserved in pestiviruses. A study showed that a mutant classical swine fever virus (CSFV) without C171 is attenuated, indicating the role of dimerization in maintaining its biological function [48]. In wild BVDV, E2 forms a disulfide-linked homodimer and also a heterodimer with E1, which functions in virus entry. Our study denotes that VLP-associated E^rns^ and E2 maintain their natural conformation, which may help maintain their immunogenicity increase specificity of the immune responses from the host.

Traditionally, VLPs are secretive and are more often purified from cell supernatant than from cell lysates. However, we found only a small portion of VLPs secreted, whereas the majority of them remained intracellularly. Whether modifications on the signal peptides can increase E2/E^rns^ secretion demands further investigation [49,50].

After in vitro verification of the VLPs generated, we next assessed the immunogenicity and the neutralization potential of them in BALB/c mice. The results showed that vaccination of mice with BVDV-VLPs induced significantly higher levels of IgG, IgG1, and IgG2a antibodies against BVDV E2 protein compared to controls in the presence of adjuvant (Figure 6A–C). In vitro neutralization assay showed that serum collected from immunized animals exhibited high-titer neutralization activity against BVDV NADL strain. Strong neutralizing activities against NADL were observed, even in the lowest concentration of VLPs tested (Figure 6D).

In addition to antibody response, we monitored cell-mediated immunity in mice immunized with BVDV-VLPs. We found both CD3^+^CD4^+^T cells and CD3^+^CD8^+^T cells were enriched upon VLP immunization (Figure 7). Moreover, mice vaccinated with the BVDV-VLPs displayed higher numbers of IL-4- and IFN-γ-secreting cells compared to mice vaccinated with the commercial BVDV-inactivated vaccine (Figure 9). Previous studies demonstrated that Th1 cytokines such as IFN-γ promote immunoglobulin class switching from IgM to IgG2a, whereas Th2 cytokines such as IL-4 lead to immunoglobulin isotype switching to IgG1 [51]. In our study, the level of E2-specific IgG1 was higher in mice immunized with BVDV-VLPs, consistent with increased production of Th2-type cytokine IL-4 in splenocytes from these mice. We were not able to determine E2-specific IgG2a due to a technical difficulty. Nevertheless, the fact that BVDV-VLP-based vaccine elicited stronger cellular immune responses than the commercial BVDV-inactivated vaccine indicates that VLP is an excellent system for the efficient induction of cell-mediated immune responses.

Taken together, our data suggest that BVDV-VLPs induce strong humoral and cellular immune responses in mice.

## 5. Conclusions

We successfully established a method to generate BVDV-VLPs with viral E2 and E^rns^ using BEVS. The VLPs produced with this protocol were morphologically and dimensionally similar to the wild-type BVDV particles. In addition, E^rns^ and E2 proteins encapsulated in the VLPs and existed in homodimers, which help preserve the immunogenicity of the two proteins. Furthermore, our BVDV-VLPs were able to induce mice to produce a similar level of humoral immune response and stronger cellular immune response than the commercial BVDV-inactivated vaccine.

## Figures and Tables

**Figure 1 vaccines-09-00350-f001:**
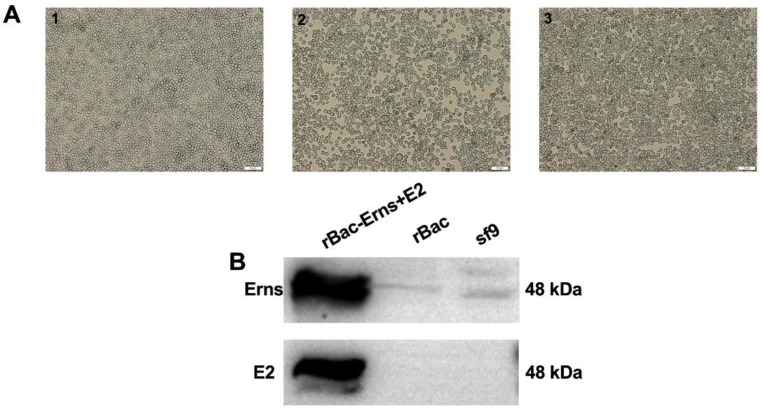
Expression of E^rns^ and E2 proteins in Sf9 Cells. (**A**) Sf9 cells mock-infected or infected with rescued recombinant viruses are shown in the following order: (1) mock-infected Sf9 cells, (2) Sf9 cells infected with rBac-E^rns^+E2 (multiplicity of infection (MOI) = 0.1), (3) Sf9 cells infected with Bac only. (**B**) Western blot analysis of E^rns^ and E2 expression in Sf9 cells. Cell lysates from Sf9 cells infected with rBac-E^rns^+E2 were subjected to Western blot analyses and E^rns^ and E2 were detected by E^rns^ pAb and E2 mAb 348, respectively. Cell lysates from Sf9 cells uninfected or infected with Bac vector were loaded as controls; numbers on the right indicate protein molecular weight in kDa.

**Figure 2 vaccines-09-00350-f002:**
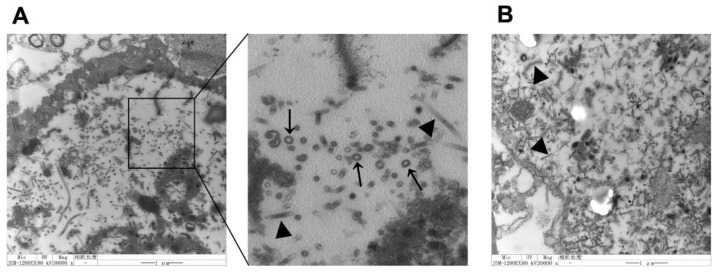
Identification of particle formation in Sf9 cells. Particle formation was confirmed by TEM in Sf9 cells infected with rBac-E^rns^+E2. Ultrathin sections of the cells showed the presence of numerous bovine viral diarrhea virus (BVDV) virus-like particles (VLPs). (**A**) Sf9 cells infected with rBac-E^rns^+E2 (MOI = 0.1). (**B**) Sf9 cells infected with Bac only. Images were obtained with a JEM-1200EX (Japan) transmission electron microscope at 80 kV. Black arrows point to VLPs, black arrowheads point to baculovirus, scale bar represents 1 μm.

**Figure 3 vaccines-09-00350-f003:**
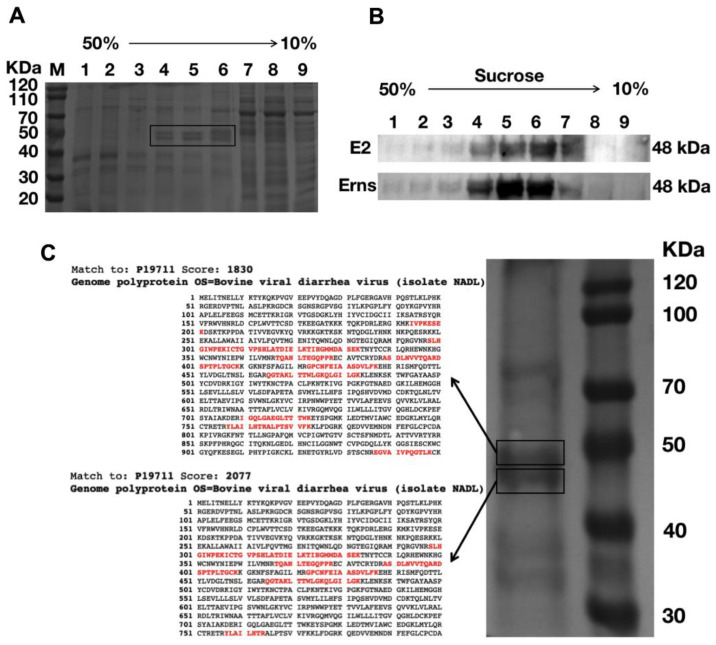
Purification of VLPs by sucrose density centrifugation. VLPs obtained with ultracentrifugation through sucrose density gradient (10%, 20%, 30%, 40%, 50%) at 35,000 rpm, 4 °C, for 3 h in a Beckman SW41 rotor and divided into nine fractions. (**A**) Fractions 1-9 were analyzed by SDS-PAGE and protein bands were visualized with Coomassie Brilliant Blue staining. The E^rns^ and E2 protein bands are highlighted in boxes; M: protein ladder. (**B**) Western blot analysis of fractions of 1-9 with E^rns^ pAb and E2 mAb 348. (**C**) Purified VLPs were subjected to SDS-PAGE and two protein bands migrated to the size of ≈48 kDa were identified by LC–MS/MS. The matched peptides are highlighted in bold red; molecular weight (kDa) of the ladder is labelled on the right.

**Figure 4 vaccines-09-00350-f004:**
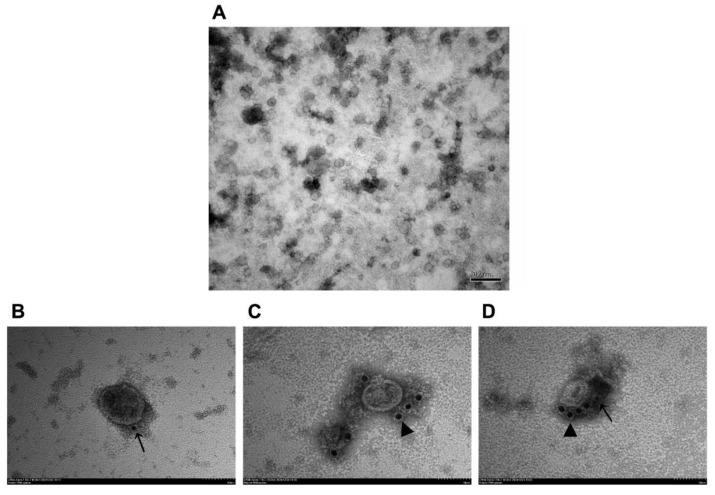
Purification TEM and immune-electron microscopy (IEM) of BVDV-VLPs. (**A**) Sucrose-purified VLPs were examined by TEM. All observed VLPs were spherical in morphology and were about 50 nm in diameter. Images were obtained with a JEM-1200EX (JEOL, Tokyo, Japan) transmission electron microscope at 80 kV; scale bar represents 200 nm. (**B**) BVDV-VLPs labeled for E2 with E2 mAb 348 and 5 nm colloidal gold-conjugated goat anti-mouse antibody. (**C**) BVDV-VLPs labeled for E^rns^ with E^rns^ pAb and 10 nm colloidal gold-conjugated goat anti-rabbit antibody. (**D**) BVDV-VLPs double-labeled for E2 (5 nm gold labels) and E^rns^ (10 nm gold labels). Black arrows point to E2, black arrowheads point to E^rns^. Images were obtained with a HT7800 (Hitachi, Tokyo, Japan) transmission electron microscope at 80 kV; scale bar represents 50 nm.

**Figure 5 vaccines-09-00350-f005:**
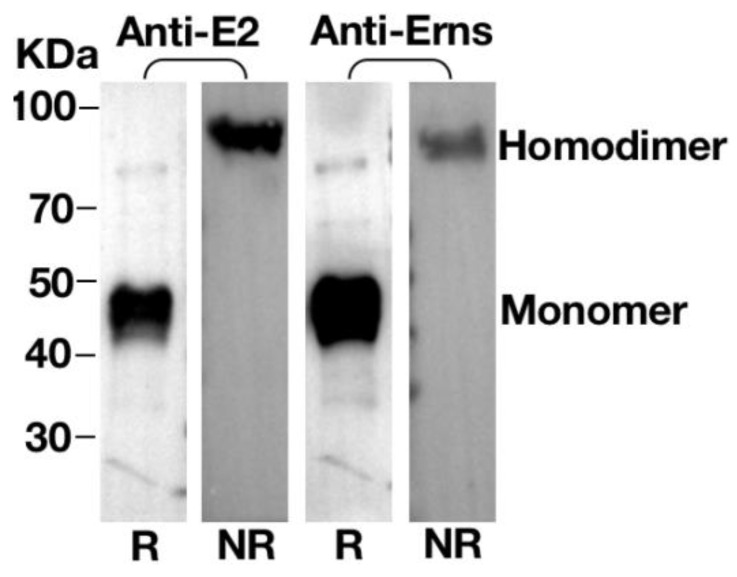
Homodimerization of E^rns^ or E2 proteins. The VLPs were separated by SDS-PAGE under reducing (R) or nonreducing (NR) conditions followed by Western blotting with E^rns^ pAb and E2 mAb 348; on the left protein ladder, the molecular weight in kDa is given. The bands of E^rns^ or E2 monomer and E^rns^ or E2 homodimer are indicated.

**Figure 6 vaccines-09-00350-f006:**
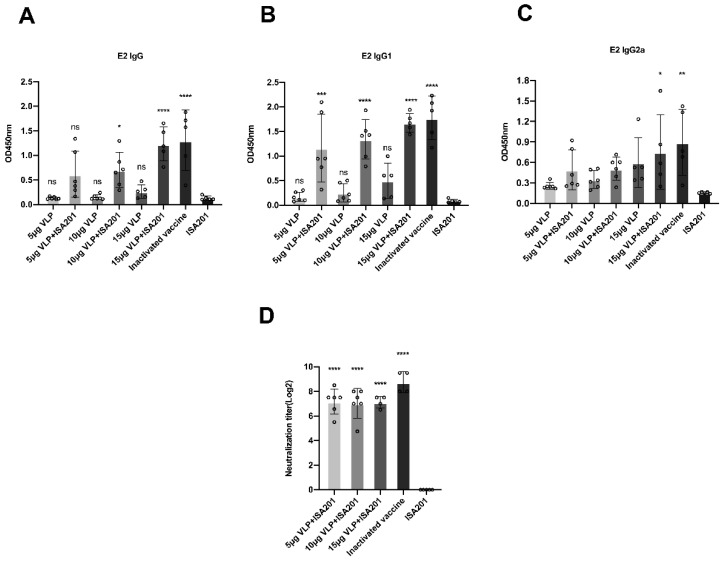
BVDV E2-specific IgG, IgG1, IgG2a and neutralizing antibodies induced by BVDV-VLPs in mice. The mice were immunized with 5, 10, or 15 μg VLPs in the presence or absence of ISA201 adjuvant; a commercial BVDV-inactivated vaccine and ISA201 adjuvant alone were used as controls. Samples were administered intramuscularly and were given twice within a 21-day interval. Serum was collected on day 42 post-prime immunization, and BVDV E2-specific IgG (**A**), IgG1 (**B**), IgG2a (**C**), and neutralizing antibody (**D**) were monitored. Error bars are standard error of the mean. GP:0.1234 (ns), 0.0332 (*), 0.0021 (**), 0.0002 (***), <0.0001 (****).

**Figure 7 vaccines-09-00350-f007:**
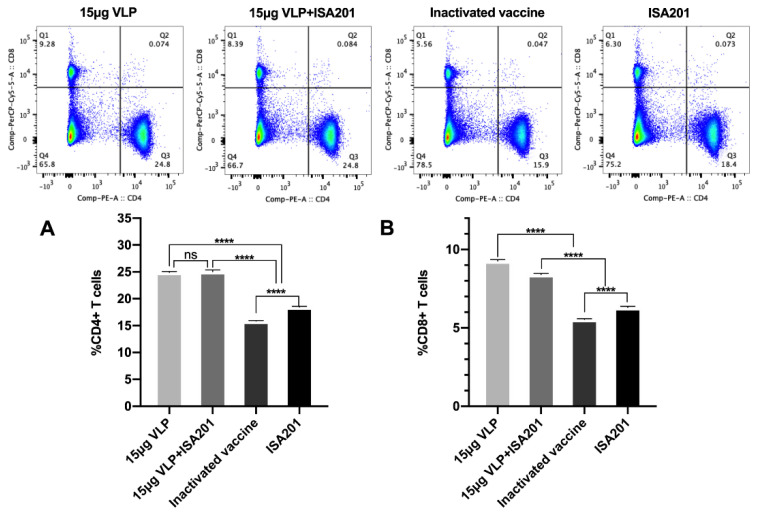
The ratio of CD3^+^CD4^+^T cells and CD3^+^CD8^+^T cells in splenocytes. Mice were immunized with 15 μg VLPs with or without ISA201 adjuvant, commercial BVDV inactivated vaccine, or ISA201 adjuvant alone; splenocytes were collected on day 28 post-prime immunization and re-stimulated with the same antigen as those used for immunization, and flow cytometry was used to detect the ratio of CD3^+^CD4^+^T cells (**A**) and CD3^+^CD8^+^T cells (**B**). Error bars are standard error of the mean. GP:0.1234 (ns), <0.0001 (****).

**Figure 8 vaccines-09-00350-f008:**
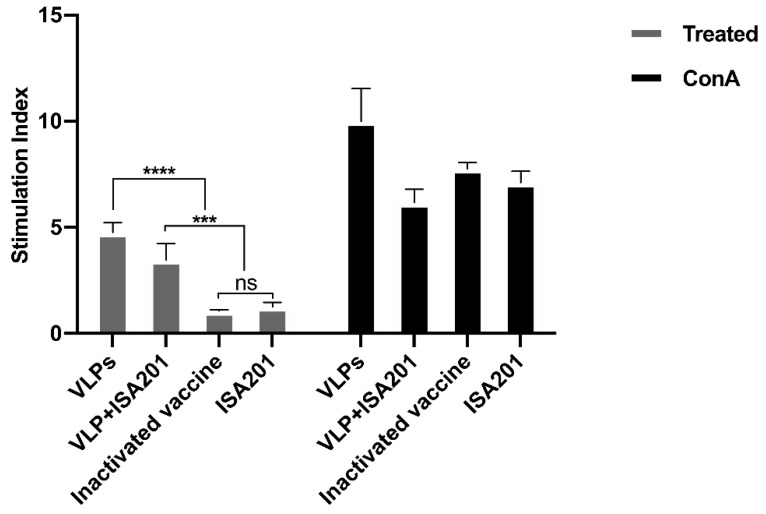
The effects of VLPs on splenocyte proliferation. Mice were immunized with 15 μg of VLPs with or without ISA201 adjuvant, a commercial BVDV-inactivated vaccine, or the ISA201 adjuvant alone. Splenocytes were collected on day 28 post-prime immunization and re-stimulated with the same antigen as those used for immunization, and cell proliferation was presented as mean of stimulation index (SI). Error bars are standard error of the mean. GP:0.1234 (ns), 0.0002 (***), <0.0001 (****).

**Figure 9 vaccines-09-00350-f009:**
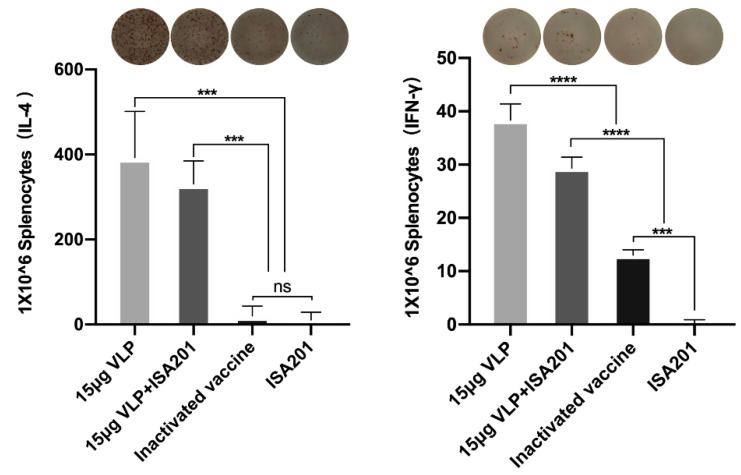
Immunization with VLPs induced IL-4 and IFN-γ secretion. Mice were immunized with 15 μg of VLPs with or without ISA201 adjuvant, commercial BVDV-inactivated vaccine, or ISA201 adjuvant alone; splenocytes were collected on day 28 post-prime immunization and re-stimulated with the same antigen as those used for immunization. Data are presented as number of IL-4- and IFN-γ-secreting cells per 10^6^ splenocytes. Error bars are standard error of the mean. GP:0.1234 (ns), 0.0002 (***), <0.0001 (****).

## Data Availability

The data presented in this study are available within the article and supplementary material.

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
