# Peer review of "Induction of Robust and Specific Humoral and Cellular Immune Responses by Bovine Viral Diarrhea Virus Virus-Like Particles (BVDV-VLPs) Engineered with Baculovirus Expression Vector System"

_vaccines, 2021, doi:10.3390/vaccines9040350_

Round 1
Reviewer 1 Report
The research result is helpful for the development of a novel vaccine against BVDV infection, in the manuscript were two questions need to explain.
- The authors need to explain the reason. In the line 97-line 108, the paragraph describes the design method that was co-expressed E0 (271-497 aa) and E2 (693-1036 aa) proteins. The predicted size of two recombinant proteins does not match with results in Fig 1B and Fig 3.
- The authors should provide further data to verify the authenticity of VLPs. It suggests that the “virus-like particles” change to recombinant XXXX in the title if the authors could not show a reasonable answer for the comment.
In Fig 4 B-D, the figures show three images of “BVDV-VLPs” is to provide the research result forming the “VLPs, however, we observed morphology of VLPs which doesn't seem particularly typical, and each image just shows single “VLPs”. It suggests providing the typical the image where have some VLPs.
Author Response
Response to Reviewer 1 Comments
Point 1: The authors need to explain the reason. In the line 97-line 108, the paragraph describes the design method that was co-expressed E0 (271-497 aa) and E2 (693-1036 aa) proteins. The predicted size of two recombinant proteins does not match with results in Fig 1B and Fig 3.
Response 1:
Thank you for you careful work. In insect cells, Erns undergoes post-translational modification and is heavily glycosylated, then its molecular weight becomes larger, which is consistent with previous reports:
[1] Munir Iqbal et al. Role for bovine viral diarrhea virus Erns glycoprotein in the control of activation of beta interferon by double-stranded RNA, J Virol. 2004 Jan;78(1):136-45. doi: 10.1128/jvi.78.1.136-145.2004
[2] M Iqbal et al. Interactions of bovine viral diarrhoea virus glycoprotein E(rns) with cell surface glycosaminoglycans, J Gen Virol. 2000 Feb;81(Pt 2):451-9. doi: 10.1099/0022-1317-81-2-451
[3] Balaje Vijayaraghavan et al. Evaluation of envelope glycoprotein E(rns) of an atypical bovine pestivirus as antigen in a microsphere immunoassay for the detection of antibodies against bovine viral diarrhea virus 1 and atypical bovine pestivirus, J Virol Methods. 2012 Nov;185(2):193-8. doi: 10.1016/j.jviromet.2012.06.017
The expression of E2 protein in this study is also consistent with previous studies:
[1] Yue Li et al. Crystal structure of glycoprotein E2 from bovine viral diarrhea virus, Proc Natl Acad Sci U S A. 2013 Apr 23;110(17):6805-10. doi: 10.1073/pnas.1300524110
[2] A S Asfor et al. Recombinant pestivirus E2 glycoproteins prevent viral attachment to permissive and non permissive cells with different efficiency, Virus Res. 2014 Aug 30;189:147-57. doi:10.1016/j.virusres.2014.05.016
Point 2: The authors should provide further data to verify the authenticity of VLPs. It suggests that the “virus-like particles” change to recombinant XXXX in the title if the authors could not show a reasonable answer for the comment.
In Fig 4 B-D, the figures show three images of “BVDV-VLPs” is to provide the research result forming the “VLPs, however, we observed morphology of VLPs which doesn't seem particularly typical, and each image just shows single “VLPs”. It suggests providing the typical the image where have some VLPs.
Response 2:
In this study, we proved that BVDV-VLPs were assembled with Erns and E2 proteins through three TEM experiments. We can provide additional electron microscope pictures to support our conclusion.
- We observed a large number of particles in insect cells with TEM(S1),We suggest you zoom in the picture。
- We purified these particles by sucrose density gradient centrifugation, and used LC-MS/MS to identify that the purified particles mainly contained BVDV Erns and E2 peptides (Figure 3C), and then we observed the purified particles are similar in size and shape to wild BVDV particles by TEM, so we determined that these particles are BVDV-VLPs(S2).
- Next, we used IEM to confirm that BVDV-VLPs are assembled from Erns and E2(S3):
Reviewer 2 Report
This manuscript addressed the production of BVDV-VLPs composed of this virus’ surface protein (Erna and E2) and their capacity to serve as vaccine, through immunization studies performed in mice and further evaluation of IgG titers and serum neutralizing capacity. The comments to this manuscript are described below:
Question 1: Since the constructs include a signal (gp64), why is most of the content in VLPs found in the intracellular compartment rather than extracellular?
Question 2: How does the antigen load (ug Erna and E2) from the VLPs compare with that of the control vaccine, in the immunization studies? Different immune responses, as well as neutralizing capacity, is presented and compared with the results using the vaccine, however without knowing the quantity of antigen used for the vaccination with the vaccine it is difficult to compare vaccine vs. VLPS. Please include this information in the manuscript.
Question 3: Why was it decided to generate VLPs based on these surface proteins only, without the use of the capsid protein ( C )? Would VLPs composed of the capsid (for structure) displaying the surface protein (for antigen display) not mimic better the viral particle?
Question 4: BVDV wild-type particles are said to be 50-80nm (line 259). The generated VLPs are also said to be in the same size range. Since VLPs are only composed of the Erna and E2 protein, is this expected, considering native viruses comprise a more complex organization with eleven functional proteins? If so, what is the structure/organization of these surface proteins in these particles in order to form round-shaped particles in the same size-range as native viruses? Are these hollow (“empty”) particles?
Line 83: “observed and verified” (two words meaning the same)
Line 443/444: brackets in the line below
Line 482: is -> are
Author Response
Response to Reviewer 2 Comments
Point 1: Since the constructs include a signal (gp64), why is most of the content in VLPs found in the intracellular compartment rather than extracellular?
Response 1:
Previous studies suggest that BVDV are assembled at membranes of the ER, and transported by the membrane system of the host cell secretory pathway to the cell surface, where they are released, probably by exocytosis. In our research, we think the role of gp64 is only to guide Erns and E2 into the endoplasmic reticulum to assemble into VLPs, and does not promote the secretion of VLPs outside the cell. Maybe the signal peptide of BVDV or other signal peptides will promote the secretion of VLPs outside the cell, we need to try it.
Point 2: How does the antigen load (ug Erna and E2) from the VLPs compare with that of the control vaccine, in the immunization studies? Different immune responses, as well as neutralizing capacity, is presented and compared with the results using the vaccine, however without knowing the quantity of antigen used for the vaccination with the vaccine it is difficult to compare vaccine vs. VLPS. Please include this information in the manuscript.
Response 2:
The control vaccine is a commercial BVDV inactivated vaccine. we can only get the information that BVDV titer is greater than 10^7 TCID50/ml, and we do not know the antigen content of Erns and E2 (ug). We set up 3 doses of VLPs vaccine (5μg, 10μg and 15μg), the purpose is to determine an antigen dose that can achieve the same immune effect as the commercial inactivated vaccine.
Point 3: Why was it decided to generate VLPs based on these surface proteins only, without the use of the capsid protein ( C )? Would VLPs composed of the capsid (for structure) displaying the surface protein (for antigen display) not mimic better the viral particle?
Response 3:
Previous studies have shown that viruses in the Flaviviridae family, such as DENV, WNV, JEV and HCV, use envelope glycoprotein can be assembled into virus-like particles, without the participation of capsid protein (C). Erns and E2 are main immunogenic proteins of BVDV, so we choose Erns and E2 envelope glycoproteins to assemble VLPs.
Point 4: BVDV wild-type particles are said to be 50-80nm (line 259). The generated VLPs are also said to be in the same size range. Since VLPs are only composed of the Erna and E2 protein, is this expected, considering native viruses comprise a more complex organization with eleven functional proteins? If so, what is the structure/organization of these surface proteins in these particles in order to form round-shaped particles in the same size-range as native viruses? Are these hollow (“empty”) particles?
Response 4:
Previous studies have shown that the wild-type BVDV is assembled by three structural proteins Erns, E1 and E2, Erns and E2 are located on the surface of the virus particle, E1 is embedded inside the virus particle, and most of virus particles is about 50nm in diameter[1]. In our study, VLPs are assembled by Erns and E2, so the size and shape are similar to wild-type BVDV. From the results of TEM and IEM(Figure 2 and Figure4), most of the VLPs are about 50nm in diameter, so we decided to change the "50~80nm" to "about 50nm" in the manuscript.
We speculate that VLPs are hollow, but we are not quite sure.
[1] Callens N et al. Morphology and Molecular Composition of Purified Bovine Viral Diarrhea Virus Envelope. PLoS Pathog 2016, 12, e1005476, doi:10.1371/journal.ppat.1005476.
Point 5: Line 83: “observed and verified” (two words meaning the same)
Line 443/444: brackets in the line below
Line 482: is -> are
Response 5:
Thank you for your careful work. We have corrected these errors in the manuscript
Reviewer 3 Report
Wang et al. Induction of Robust and Specific Humoral and Cellular Immune Responses by Bovine Viral Diarrhea Virus Virus-Like Particles (BVDV-VLPs) Engineered with Baculovirus Expression Vector System
The manuscript entitled, “Induction of Robust and Specific Humoral and Cellular Immune Responses by Bovine Viral Diarrhea Virus Virus-Like Particles (BVDV-VLPs) Engineered with Baculovirus Expression Vector System” by Wang et al. describes the development of a VLP-based vaccine for BVDV infection prevalent in cattle. The authors describe the isolation of VLPs from Sf9 cells transduced by baculovirus expression vectors (BEVs) that express the BVDV proteins E2 and Erns, which are individually fused to GP64. The authors show through TEM and Western blot analysis that VLPs expressing the viral antigens can successfully form in and be isolated from Sf9 cells. Importantly, they present evidence via flow cytometry, NAb, and EliSpot assays that the VLPs can elicit a strong immune response in mice.
In this reviewer’s opinion, the results are fairly straightforward to interpret and the necessary controls are used for all experiments. The data is quite convincing and should be informative and valuable to the Infectious Disease and Agricultural Sciences communities. However, the presentation of the manuscript requires substantial editing and restructuring. For this reason, the reviewer recommends that the manuscript be accepted for publication in Vaccines only following revision. The following issues that should be addressed are provide in the bullet points below.
- The authors fail to place their work in context with other relevant studies in the field. There have been many years of research using BEV/Sf9 systems for vaccine strategies against BVDVs. It would be best to reference and discuss these studies, so that the reader has a better context for the novelty in the specific approach(es) taken. A few relevant papers that come to mind are:
- Thomas et al. (2009), Evaluation of efficacy of mammalian and baculovirus expressed E2 subunit vaccine candidates to bovine viral diarrhoea virus (PMID: 19428855)
- Pande et al. (2005), “The glycosylation pattern of baculovirus expressed envelope protein E2 affects its ability to prevent infection with bovine viral diarrhoea virus” (PMID: 15993973)
- Chimeno Zoth et al. (2007), “Recombinant E2 glycoprotein of bovine viral diarrhea virus induces a solid humoral neutralizing immune response but fails to confer total protection in cattle” (PMID: 17581680)
- The reviewer appreciates that the authors are not native English speakers/writers. However, there are many grammatical mistakes throughout the manuscript - too many to mention here. The paper should be edited for grammar before resubmission.
- In many instances, the figure legend texts are nearly word-for-word repeated from the main text. Please revise the writing to reduce the frequency of redundant descriptions.
- Similarly, the Discussion is a rehash of the findings described in the results. A better use of this section is to discuss the context of the work related to previous attempts in engineering BEV-based vaccines for BVDV.
- In Figure 5, the authors use the term “Monodimer”. Did they mean “monomer”?
- In many parts of the manuscript, the authors use the phrase “CD3+CD4+ T cells and CD3+CD4+ T cells”, which is the same, leaves out mention of CD8+ T cells, and is likely a typo. Please fix these instances:
- Page 10, lines 357-358
- Page 11, lines 364 and 368
- Page 13, lines 465
- Can the authors provide discussion on the percentage of VLPs that express E2 and Erns following purification?
Author Response
Response to Reviewer 3 Comments
Point 1: The authors fail to place their work in context with other relevant studies in the field. There have been many years of research using BEV/Sf9 systems for vaccine strategies against BVDVs. It would be best to reference and discuss these studies, so that the reader has a better context for the novelty in the specific approach(es) taken. A few relevant papers that come to mind are:
Thomas et al. (2009), Evaluation of efficacy of mammalian and baculovirus expressed E2 subunit vaccine candidates to bovine viral diarrhoea virus (PMID: 19428855)
Pande et al. (2005), “The glycosylation pattern of baculovirus expressed envelope protein E2 affects its ability to prevent infection with bovine viral diarrhoea virus” (PMID: 15993973)
Chimeno Zoth et al. (2007), “Recombinant E2 glycoprotein of bovine viral diarrhea virus induces a solid humoral neutralizing immune response but fails to confer total protection in cattle” (PMID: 17581680)
Response 1:
Thank you for your valuable advice, in the discussion section, we have compare our study with previous studies in the field.
Point 2: The reviewer appreciates that the authors are not native English speakers/writers. However, there are many grammatical mistakes throughout the manuscript - too many to mention here. The paper should be edited for grammar before resubmission.
Point 3: In many instances, the figure legend texts are nearly word-for-word repeated from the main text. Please revise the writing to reduce the frequency of redundant descriptions.
Similarly, the Discussion is a rehash of the findings described in the results. A better use of this section is to discuss the context of the work related to previous attempts in engineering BEV-based vaccines for BVDV.
Response 2 and Response 3:
We will correct the grammatical errors and enrich the discussion content in the manuscript.
Point 4: In Figure 5, the authors use the term “Monodimer”. Did they mean “monomer”?
In many parts of the manuscript, the authors use the phrase “CD3+CD4+ T cells and CD3+CD4+ T cells”, which is the same, leaves out mention of CD8+ T cells, and is likely a typo. Please fix these instances:
Page 10, lines 357-358
Page 11, lines 364 and 368
Page 13, lines 465
Can the authors provide discussion on the percentage of VLPs that express E2 and Erns following purification?
Response 4:
Thank you for your careful work. I have changed "Monodimer" to "monomer", and changed“CD3+CD4+ T cells and CD3+CD4+ T cells”to“CD3+CD4+ T cells and CD3+CD8+ T cells” in this manuscript.
We are currently unable to detect the percentage of VLPs that express E2 and Erns following purification, but a new study in our laboratory is working to establish a double-antibody sandwich Elisa method to detect the amount of E2 and Erns in VLPs.